# Escape Survival and Scale Damage Assessment of Red Mullet (*Mullus barbatus* Linnaeus, 1758) during Bottom Trawling in the Central Mediterranean Sea

**DOI:** 10.3390/biology12050649

**Published:** 2023-04-25

**Authors:** Michele Luca Geraci, Giacomo Sardo, Fabio Falsone, Danilo Scannella, Michael Breen, Fabio Fiorentino, Antonello Sala, Sergio Vitale

**Affiliations:** 1Institute for Marine Biological Resources and Biotechnology (IRBIM), National Research Council–CNR, 98122 Mazara del Vallo, TP, Italy; 2Department of Biological, Geological and Environmental Sciences (BiGeA)–Marine Biology and Fisheries Laboratory of Fano (PU), University of Bologna, 40126 Bologna, BO, Italy; 3Institute of Marine Research, NO-5817 Bergen, Norway; 4SZN, Lungomare Cristoforo Colombo 4521, 90149 Palermo, PA, Italy; 5Institute for Marine Biological Resources and Biotechnology (IRBIM), National Research Council–CNR, 60125 Ancona, AN, Italy

**Keywords:** fishing mortality, trawl fishery, Strait of Sicily, MCRS

## Abstract

**Simple Summary:**

Fisheries are amongst the most important anthropogenic activities that strongly impact the marine environment. Therefore, stock assessments are routinely carried out to evaluate the status of the main commercial species to avoid overexploitation. However, these assessments do not take into account the possible mortality of fish due to interaction with fishing gear, and the evaluation of escape survival rates is needed. For the first time in the Central Mediterranean, this study evaluated the escape survival of the red mullet escaping from a bottom trawl. The survival rate of the control individuals (open codend) was higher compared to the treatment individuals (closed codend). Larger fish in the treatment group had a higher probability of dying, while the opposite was observed in the controls. In addition, treatment fish had significantly more wounds per fish than control fish, and were mainly injured around the head. The results highlighted that the sampling methodology was effective in collecting fish samples without affecting their survival rates. Although promising, these results need to be confirmed by further investigation before applying the survival rates in stock assessment models.

**Abstract:**

Stock assessments routinely evaluate the status of commercially harvested species, but seldom account for the possible mortality of released or escaping fish. This study presents a method for estimating the escape survival of the red mullet (*Mullus barbatus*) from demersal trawling in the Central Mediterranean Sea. Fish escaping from the trawl codend were collected in a detachable cage, which was lined to reduce water flow and protect the sampled fish from further fatigue and injury. Control fish (from an open codend) showed high survival, 94% (87–97%, 95% Confidence Interval), and minimal injuries, while fish escaping through codend meshes had significantly increased injuries and reduced survival, 63% (55–70%). During 7 days of captive monitoring, treatment group mortality was highest in the first 24 h and ceased for both groups within 48 h. Conflicting length-related mortality was observed, where larger treatment fish had a higher probability of dying, while the opposite was observed in the controls. Analysis showed that treatment fish were significantly more injured than control fish, with treatment fish predominantly injured in the head zone. In conclusion, this improved methodology should be repeated to provide accurate escape mortality estimates for the improved stock assessment of the red mullet in the Central Mediterranean.

## 1. Introduction

Fishing activity is one of the most important anthropogenic impacts affecting marine resources and consequently the ecological equilibria in marine ecosystems [1,2,3,4]. Among the different fisheries, bottom trawling is considered one of the most impactful techniques in the Mediterranean due to high quantities of unwanted catch, including the incidental catch of non-target species and juveniles of both target and non-target species, which are either discarded because of their low economic value or because of legal restrictions, e.g., [5,6,7]. Excluding temporal and spatial closures, the most common way of addressing the issue of by-catch in towed gear has been to improve gear selectivity. Indeed, to mitigate the problem, the European Union established conservation measures such as minimum legal mesh size and the Minimum Conservation Reference Size (MCRS), under Reg. EU 1241/2019 [8], while Reg. EU 1380/2013 [9] encourages improving the selectivity of the fishing gear. Therefore, in recent years, physical modifications such as by-catch reduction devices (BRDs) have been tested to improve species selection, e.g., [10,11,12,13,14]. In some cases, compared to conventional gear, these latter sorts of BRDs have been demonstrated to mitigate problematic by-catches [10,11,15,16]. However, the survival rates of escaping fish, either from a traditional codend or from gear with a BRD, are not well-known, despite their potential to bias stock assessment estimates of yield and biomass [17].

Although stock assessments generally assume all escaping fish survive and grow, several experiments have shown that individuals escaping from nets during fishing may not always survive [18]. The objective of studies on escape mortality is to demonstrate that a sufficient proportion of the released fish survive to justify their short-term loss from the catch. In many cases, escape occurs after the fish have been subjected to a wide variety of capture-related stressors and possible injury through contact with other fish, debris, or the fishing gear itself (reviewed by [18,19,20]). For trawled gear, the key variables affecting escape survival include tow duration, catch composition, the weight of the catch, mesh size/shape, and the codend circumference [18,21,22,23,24]. In addition, the survival rates are species-specific; for example, Atlantic cod (*Gadus morhua*) has been found to be more resistant than other gadoids with respect to escape mortality from demersal trawls. Scale damage is a common injury of fish caught in trawled gear but not in hook fishing [25]. 

In Atlantic waters, initial works by Main and Sangster [26], Main and Sangster [27] and Main and Sangster [28] suggested that survival rates for haddock (*Melanogrammus aeglefinus*) and whiting (*Merlangius merlangus*) could be quite high (80–90%), depending on the mesh size used. Soldal et al. [25] reported that mortality was less than 10% for haddock and 0% for Atlantic cod escaping from a demersal trawl. Robinson et al. [29], studying the survival rates of groundfish such as Atlantic cod, American plaice (*Hippoglossoides platessoides*), and yellowtail flounder (*Myzopsetta ferruginea*), concluded that survival rates were extremely variable and dependent on the season. Therefore, overall, the results of the studies on escape mortality suggest that the mortality associated with capture and escape may be relatively low for many species, particularly for gadoids and flatfishes. However, it is also obvious that not all fish survive the process of capture and escape. Measuring the survival of fish escaping from fishing gear under various fishing conditions is not an easy task. It is subject to high variability and methodological flaws. It is therefore not surprising that the accuracy of the escape mortalities estimated in various studies has been criticized.

In the Mediterranean Sea, to date, the few studies carried out to assess escape survival have only been conducted in the Aegean Sea (eastern Mediterranean) [30,31,32,33,34,35,36,37,38]. In particular, Metin et al. [30] found that the survival rate of red mullet (*Mullus barbatus)* after escaping from a commercial bottom-trawl codend (40 mm) was on average 93%. More recently, Düzbastilar et al. [37] found a seasonal effect on the survival of *M. barbatus*, with mean escape mortality significantly higher in winter compared to summer. Furthermore, mortality was also highest among the smallest fish, particularly during winter. In the present study, the survival rate of red mullet escaping through a commercial trawl codend was estimated for the first time in the Central Mediterranean Sea.

## 2. Materials and Methods

### 2.1. Study Area

In July 2018, a study to estimate the escape survival of the *M. barbatus* was carried out in the Strait of Sicily, located in the Central Mediterranean Sea and separating the eastern and the western basins (Figure 1). 

Peculiar oceanographic features contribute to making the Strait of Sicily one of the most productive areas for demersal fisheries in the Mediterranean, exploited mainly by trawl fisheries [39,40]. In particular, the trawling fleet exploits the deeper fishing ground, targeting deep-water crustaceans such as *Aristaeomorpha foliacea*, *Aristaeus antennatus*, *Parapenaeus longirostris*, and *Nephrops norvegicus* [41,42] as well as the shallow fishing ground, targeting mainly *M. barbatus*, *M. surmuletus*, sparids, and cephalopods [43,44,45].

### 2.2. Overview of the Scientific Approach

The approach used to assess the escape survival and the scale damage was based on a comparison of the state of fish caught by two configurations of trawl net, one with an open codend and another with a closed codend, in which escaping fish had to pass through the codend meshes. In both configurations, the codend was covered by a flexible cover that retained fish escaping from the codend in a detachable cage. The adopted procedure had three steps. The first step included the design and preparation of the experimental gear (cage—control and treatment, cover, and liner), the protocols of the manipulation of fish at sea as well as the scale damage and injury assessment protocols. The second step involved the placement of the cages at an inshore site and the monitoring of the fish sampled. Lastly, during the third step, the data were collected and analyzed (Figure 2).

### 2.3. Cage Description 

To collect samples either from the treatment (closed codend) or control (open codend) hauls, a detachable cage (two identical, one for control and one for treatment) with a nominal mesh size of 20 mm square was attached to a fixed cover, in turn fixed to the forward end of the codend. The structure was stabilized using 5 rigid hoops of 2 m diameter. The cages were about 10 m long and rigged with two 1.0 m long horizontal zippers to retrieve any dead fish and feed the survivors. Based on Breen et al. [46], a liner was installed at the end of the codend cover and cage assembly to protect the fish in the cage from excessive water flow (Figure 3). This cage liner was removed after the cage was detached from the codend cover.

### 2.4. Sampling at Sea 

The experiment was conducted between latitude 37°31.690–37°32.780 N and longitude 12°38.730–12°36.450 E, at a mean depth of 50 m (Figure 1). Two hauls were carried out on the same day, 18 July 2018; one of these, the treatment, was conducted with a closed commercial codend (40 mm square) according to Reg. EU 1241/2019 (see Geraci et al. [16] for a more detailed description of the gear) whereas during the control haul the codend was kept open (Figure 4). 

Haul duration was fixed at 30 and 15 min, respectively, for treatment and control, with a mean towing speed of 2.8 knots. The main reason for limiting the towing duration of the control to 15 min was to avoid over-filling the sampling cages with fish. The difference in towing duration between the treatment and control should have a minimal effect on the samples of escaped fish because the cage lining specifically protects them from fatigue after escape.

Two professional divers were employed during the sampling activity. At the end of each haul, a speed of 0.2 knots was maintained in order to avoid the collapse of the cage. Then, the two divers descended to the cage (about 50 m); one held the cover while the other removed the cage from the liner (similar to removing a “sock”). After that, the divers detached the cage and gradually raised it up to about 10 m depth in 20 min. When the cage reached the predetermined depth (10 m), floats were attached to maintain the cage at this depth. Finally, a rubber dinghy was used to tow the cage, at a speed of 0.2 knots, to the inshore monitoring site, where it was anchored to the seabed. The same methodologies were repeated for each haul/cage (treatment and control).

### 2.5. Monitoring of Fish Survival

The cages for housing the sampled fish were anchored at depths of about 10 m (37°34.041 N; 12°39.148 E). The shallow depth was chosen to enable the divers to operate at safe depths to feed fish, extract dead individuals, and assess the vitality of survivors. The temperature at the fishing ground was 18 °C while in the monitoring site it was 20 °C. Fish were observed by divers three times a day (“early” at 09:00, “middle” at 14:00, and “late” at 19:00) over a period of 7 days (from 18 to 25 July 2018). After 2 days, they were fed with shrimps, worms, and feed pellets. In addition, the behavior of the captive fish was monitored and recorded by the divers using the same underwater camera systems used by Sardo et al. [47] (i.e., GoPro Hero 4, GmbH, München, Deutschland) (Figure 5). Given that the wild fish were kept in captivity, their behavior could be affected during the monitoring period. All surviving individuals were released into the sea 7 days after capture. 

### 2.6. Scale Damage and Injury Analysis

The *M. barbatus* specimens that died during the monitoring period were assessed for injuries and scale damage, whereas all surviving individuals were released into the sea at the end of the experiment. Dead specimens were photographed using the above-mentioned video camera mounted on an external tripod 3-Way 2.0 with an extension arm. The video camera was positioned at a height of 50 cm from the ichthyometer and pictures for both the right and left sides of each specimen were taken. During image processing, a grid-square was applied to the pictures to assess injuries to the specimens in each of the 4 zones: head, abdominal, anal, and caudal. Specifically, the head zone went from the mouth to the operculum, the abdominal zone from the operculum up to halfway between the tip of the pectoral fin and the origin of the anal fin, the anal zone from halfway between the tip of the pectoral fin and the origin of the anal fin to the insertion of the dorsal fin, and the caudal zone from the insertion of the dorsal fin to the origin of the caudal fin (Figure 6). A total of 118 pictures were analyzed with the help of ImageJ2 software [48].

In turn, each zone was further sub-divided into several equally distributed sub-squares. Four authors of the present study, M.L.G., G.S., F.Fa, and D.S., counted independently the number of squares per zone where scale damage and injuries were visible on the total number of squares covered by the body surface of each individual per zone, excluding the eyes and the fins. Therefore, the scale damage and the injuries were quantified as a proportion by body zone.

### 2.7. Survival Analysis

The sizes of the surviving individuals were estimated through ImageJ2 software [48] taking video frames from the GoPro 4 Hero black. The distance between two consecutive hoops in the cage (1 m) was used as a standard. Dead specimens were measured (Total Length, TL) and weighed (W) to the nearest 5 mm and 0.1 g, respectively. 

All the analyses and graphical representations were carried out through R version 4.2.2 [49]. The survival rates of the treatment and control individuals were calculated as a percentage, namely the number of survivors at a given time divided by the total number of individuals at the beginning of the experiment. The Wilson method [50] was applied to calculate the 95% confidence interval for these estimates using the R package *binom* [51] as recommended by the ICES WKMEDS (International Council for the Exploration of the Sea—Workshop on Methods for Estimating Discard Survival) [52]. 

The probability of survival over time was estimated using the non-parametric Kaplan–Meier function applied to both treatment and control groups, with the R packages *binom* [51], *survminer* [53], and *survival* [54], where time zero was hauling time for each haul and surviving individuals were right censored. 

The relationship between survival and individual length (in 5 mm length classes) was investigated using a generalized linear model (GLM) [55], fitted using the binomial error distribution and logit-link functions. In the survival model, the survival proportion was used as a response variable while TL, the haul type (2 levels: treatment and control), and the interaction between them were used as predictors. The significance of the model terms was assessed using likelihood ratio testing. 

The scale damage and injury data were assessed by fitting a Generalized Linear Mixed Model (GLMM). In the full model, the proportion of damage was the response variable and individual size, haul type (2 levels: treatment and control), body zone (4 levels: head, abdominal, anal, caudal), and the interaction of haul type and zone were used as predictors. All models were fitted with beta error distribution (logit link function) using the R package *glmmTMB* [56] and included a random intercept term for individual fish to account for autocorrelation between body zones on the same fish. Given that the beta distribution is bounded by the open interval 0 to 1, (i.e., excluding 0 and 1), the proportions of damage were transformed in accordance with the Smithson and Verkuilen [57] methods. The model’s residual patterns were checked for the violation of assumptions using the R package *DHARMa* [58]. Residual homoscedasticity in the models was addressed using variance structures to account for differing residual dispersion in the haul types and body zone groups [56,59]. The significance of model terms was determined using likelihood ratio testing. The 95% confidence interval (CI) was estimated through the t-student distribution of the model prediction. The relationship between the proportion of damage, by body zone, and total length was presented using a linear model with splines (three degrees of freedom) with *ggplot2* [60] and *splines2* [61] packages.

## 3. Results

In total, 256 individuals belonging to 11 species were caught in the course of the experiments. The survival rate for the control samples of red mullet (i.e., escaping through an open codend) was high. By the end of the monitoring period, 54 and 5 red mullets had died from treatment and control hauls, respectively (Table 1). In addition, *Arnoglossus thori*, *Bothus podas*, *Dactilopterus volitans*, *Pagellus acarne*, *P. erythrinus*, *Serranus cabrilla*, *S. hepatus*, *Spicara flexuosa*, and *Synodus saurus* also appeared to have high survival rates but due to small sample sizes (n < 10) were not included in further analyses (Table 1).

Similar to the behavior described by Metin et al. [30], immediately after the detachment of the cage from the trawl *M. barbatus* individuals were resting with high operculum activity on the bottom. During the first observations, a few hours after anchoring the cages, red mullets were swimming actively in a shoal. Some species such as *P. erythrinus* also swam actively with the red mullet shoal whereas others such as *S. saurus* remained stationary on the bottom of the cage. 

The observed survival rate of *M. barbatus* sampled during the treatment haul was 63% (55–70% upper and lower bounds of the 95% CI), while the control sample had a significantly higher survival of 94% (87–97% CI). The Kaplan–Meier plot (Figure 7) shows that most treatment specimens died between 12 and 24 h, after which the mortality rate slowed slightly and ceased after 48 h. The control mortality occurred between 24 and 48 h at a significantly lower rate, also ceasing after 48 h (Figure 7).

The size distributions of the individuals caught during treatment and control hauls were mainly composed of fish larger than the MCRS, as defined by Reg. EU 1241/2019 (Figure 8).

The survival of red mullets in the treatment group (i.e., escaping through codend meshes) was inversely related to total length, with larger fish having a lower probability of surviving (Figure 9). Conversely, in the control group, the data suggest there may have been a lower probability of survival in the smallest fish (between 100 and 125 mm TL), although this is based on a small number of dead fish (five individuals) (Figure 9).

The GLM of the survival data confirmed that the treatment group had a significantly lower survival than the controls (Table 2). It also confirmed that total length also significantly affected survival, with a highly significant interaction between the treatment and control groups, as described in Figure 9.

The proportion of damage was significantly higher in red mullets from the treatment group (Figure 10, Table 3). Moreover, the treatment fish had significantly more damage to the head (Figure 10). In the control fish, the damage was highest in the abdominal zone, but this was not significant compared to the other zones. The effect size on damage was not significant in the GLMM and so was excluded from the final model (Table 3). The residuals in the final model were marginally under-dispersed. This implies that inferences about the significance of the effects were marginally conservative, with an increased likelihood of a type-II error (i.e., false negative). The random variation for individual fish was low (standard deviation = 0.451), indicating that observed damage was proportionate across all zones within individual fish. No barotrauma-related injuries were observed.

The proportion of skin damage in the head zone of the red mullet from the treatment group was lowest in the smallest fish, increasing between 120 and 130 mm TL to a maximum of approximately 0.3 at 140 mm TL, after which it remained approximately constant (Figure 11). Damage in the other zones was consistently low across the size range.

## 4. Discussion

The survival rate of red mullet (*M. barbatus*) escaping from a commercial trawl codend (40 mm square) was evaluated for the first time in the Central Mediterranean. Overall, the survival rate of *M. barbatus* was high in the treatment group (63%; 55–70% CI) and very high in the control (94%; 87–97% CI). All observed mortality occurred within the first 2 days, which agrees with previous studies reporting that mortalities occur during the first 2–3 days [30,62,63] and suggests that the monitoring period (7 days) is adequate to estimate the survival rates of the red mullet escaping from a trawl codend. 

The survival rate of the red mullet sampled in the present study (63%; 55–70% CI) from the treatment haul was lower than that reported in the Aegean Sea. In particular, Metin et al. [30] reported an average of 93% survival for red mullet escaping from a square mesh 40 mm PE codend in the experiments conducted in September, whereas Duzbastillar et al. [32], using the same codend, found a 95.1% mean mortality in October. In addition, Duzbastillar et al. [32] also tested a 40 mm diamond mesh codend, which showed lower survival rates (81.2%) than the square mesh, suggesting an effect of the mesh shape on red mullet survival. Lastly, Duzbastillar et al. [35], testing three different codends (40 mm square, 44 mm diamond, and 50 mm diamond), found survival rates of red mullet lower than the present study with the 44 mm diamond (about 54%) and higher survival rates with 40 mm (about 74%) and 50 mm (about 73%) codends. Given that our experiment was carried out in July, the higher temperature might have caused such differences. Conversely, Duzbastillar et al. [37] found higher mortality in winter from small red mullet specimens and suggested that the observed mortality may be related to the physical condition of the fish, being that fish are less nourished during winter and less able to recover from exhaustion and physical injuries during the catching process. On the other hand, red bandfish (*Cepola macrophthalma*) caught in the Aegean Sea showed a significant effect of temperature on the survival of escaped specimens [34] whereas no seasonal effects on survival were observed in common pandora (*Pagellus erythrinus*) [38]. 

Surprisingly, Duzbastillar et al. [37] and Metin et al. [30] in the Aegean Sea, using a similar sampling methodology, found that red mullet controls had significantly higher mortality than the treatment groups. The higher control survival rates (94%; 87–97% CI) observed in our experiment are likely due to sampling and handling procedures that aimed to minimize stress on the sampled fish, including minimizing barotrauma through controlled decompression and reducing additional fatigue by reducing water flow in the cover with the cage liner [46]. The very high survival of the control fish in this study provides strong evidence that the methodological approach did not adversely affect the survival of the samples of red mullet in this experiment. However, it should be noted that these observations are not properly representative of the conditions experienced during commercial fishing hauls with respect to depth, towing duration, and catch size. Mediterranean trawl fisheries typically target multiple species and operate at depths of up to 800 m [64], whereas the red mullet lives at depths of <200 m [65]. The experimental hauls in this study were carried out in shallow waters (about 50 m), as fishing in deeper waters would have exposed fish to higher decompression stress compromising their vitality, as with other Mediterranean studies [30,36,37]. Furthermore, the towing durations (15 min control and 30 min treatment) were short compared with 1–2.5 h in commercial fishing hauls (deeper than 50 m). Therefore, in order to provide valid scientific justification for the implementation of regulations on trawl codend mesh size and MCRS in the Mediterranean (as defined by Reg. EU 1241/2019), further studies using methods similar to those describe here should be repeated with sufficient replicates to provide a robust description of the natural variation in escape survival rates. These studies should also attempt to replicate variations in commercial fishing operations, including seasonality, fishing depth, towing duration, and associated catch sizes and compositions. Data from these studies could be used to reduce the uncertainty in stock assessment modeling associated with unaccounted mortality and its bias of fishing mortality estimates [17,18,19,20,66,67]. In the meantime, the sensitivity of stock assessment models to such escape mortality data could be assessed using simulation exercises (e.g., [17]). 

A significantly negative effect of total fish length (TL) on red mullet survival was observed in the treatment group in this study (Figure 9). This contrasts with observations showing the opposite effect in red mullet in the Aegean Sea [30,37] as well as several other teleost species in the Mediterranean [31,36] and Atlantic [18,33,68]. While species-specific variation in the effect of fish size upon escape mortality is recognized [18,21,33,69], the contradiction with similar studies on the same species is noteworthy. Our observation that survival decreases with increasing size in fish escaping through codend meshes better fits the hypothesis that these fish are more likely to experience fatal injuries and stresses as they struggle to pass through the codend meshes [18,25]. Furthermore, the degree of injury, particularly to the head, was significantly greater in the treatment group fish relative to the controls (which did not pass through codend meshes), further supporting this hypothesis (Table 3; Figure 10). However, evidence that injuries were also length-related was non-conclusive (Table 3; Figure 11).

Several studies have observed, using video cameras, that after escaping from the trawl codend some fish, particularly smaller ones, are unable to sustain swimming against the water flow in the cover/cage and fall back to the rear where they likely sustain further injuries [25,70,71,72]. These observations have been used to explain some length-related mortality and injury effects observed in escape survival studies [71,72], as well as to propose and demonstrate the effectiveness of reducing water flow in the sampling cover and its benefits for survival estimation [46]. The survival studies on red mullet in the Aegean Sea [21,28] used a similar method to the current study, except they did not use a liner in the cover to reduce water flow in the sample cage during towing. These and related studies observed increased mortality in the smallest fish [30,37], as well as increased injuries in the tail region [35], which better fits the hypothesis of sampling-induced mortality. This contrast with our results further supports the conclusion that the methods used in the present study have provided more reliable estimates of escape survival. 

## 5. Conclusions

This study has described an effective method for estimating escape mortality in red mullets escaping from trawl codends, in particular the use of a cover line to protect fish from injurious water flows during sampling. The observed survival rate of *M. barbatus* in the treatment group was 63% (55–70% CI), while in the control group it was 94% (87–97% CI). It is recommended that further studies using methods similar to those described here should be conducted to provide valid scientific justification for the implementation of regulations for trawl codend mesh size and MCRS in the Mediterranean (as defined by Reg. EU 1241/2019). These studies should have sufficient replicates to provide a robust description of the natural variability in escape survival rates, as well as attempt to replicate variations in commercial fishing operations including seasonality, fishing depth, towing duration, and associated catch sizes and compositions. The data resulting from this work will be essential for removing biases in fishing mortality estimates because of the inherent assumption in most stock assessment models that all escaping fish survive. Furthermore, they will enable fishery managers to make more informed decisions about the implementation of different gear selectivity scenarios based on more reliable projections of yield and biomass in the exploited populations. 

## Figures and Tables

**Figure 1 biology-12-00649-f001:**
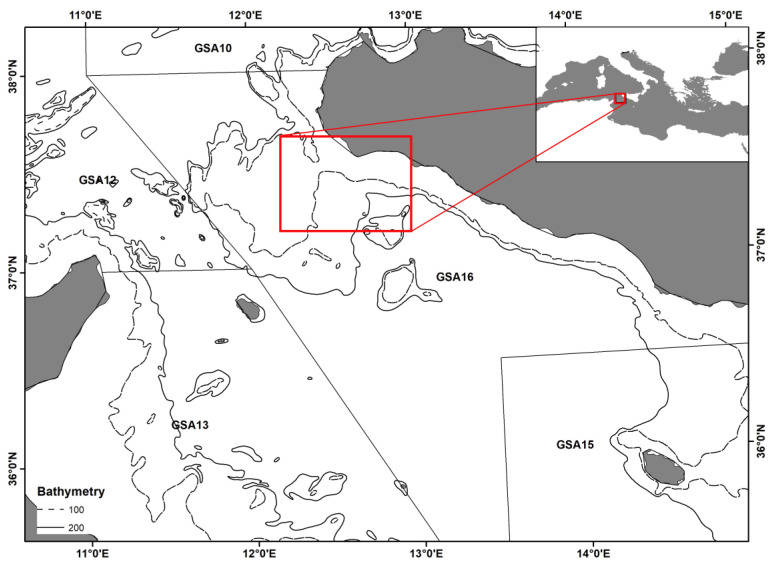
The study area is highlighted with a black square box (maps from Vitale et al. [10,11]).

**Figure 2 biology-12-00649-f002:**
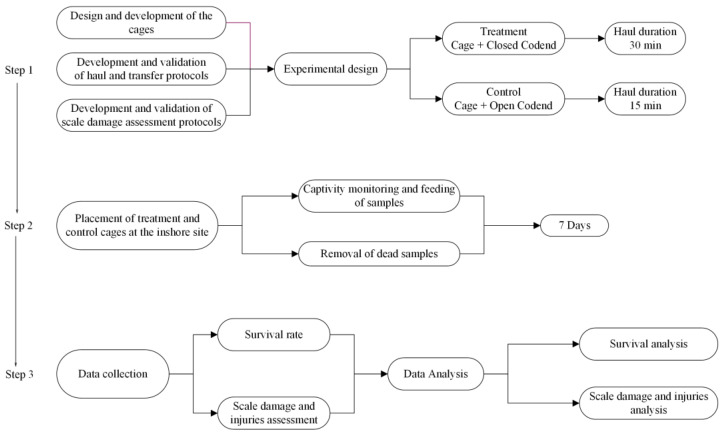
Diagram flows showing the steps followed to carry out the present study experiment.

**Figure 3 biology-12-00649-f003:**
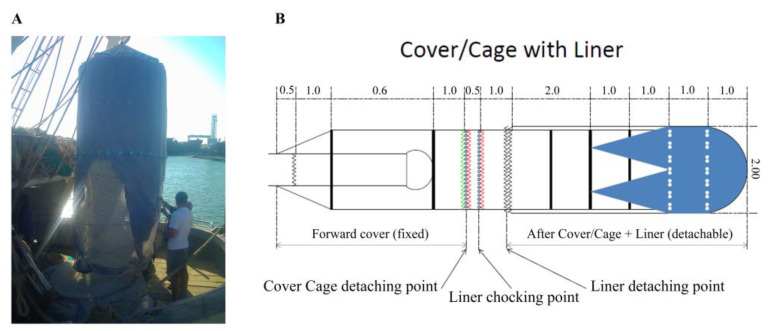
The cage with liner used during the experiment (**A**) and a schematic representation of the cover/cage with liner (**B**).

**Figure 4 biology-12-00649-f004:**
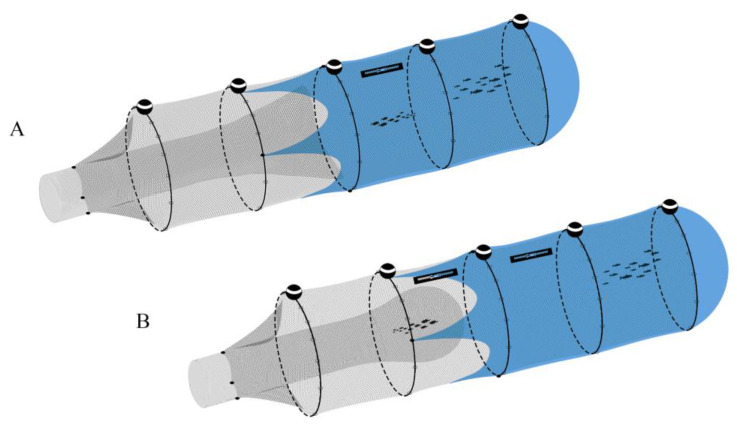
The gear used to sample red mullet individuals. (**A**) treatment, closed commercial codend; (**B**) control, opened commercial codend.

**Figure 5 biology-12-00649-f005:**
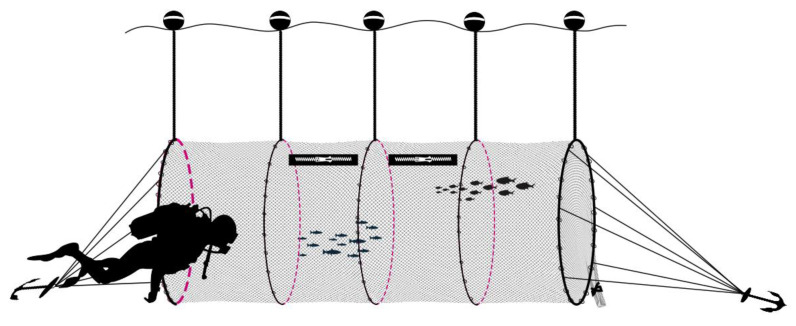
Drawing showing a diver who monitored fish condition during the captivity period.

**Figure 6 biology-12-00649-f006:**
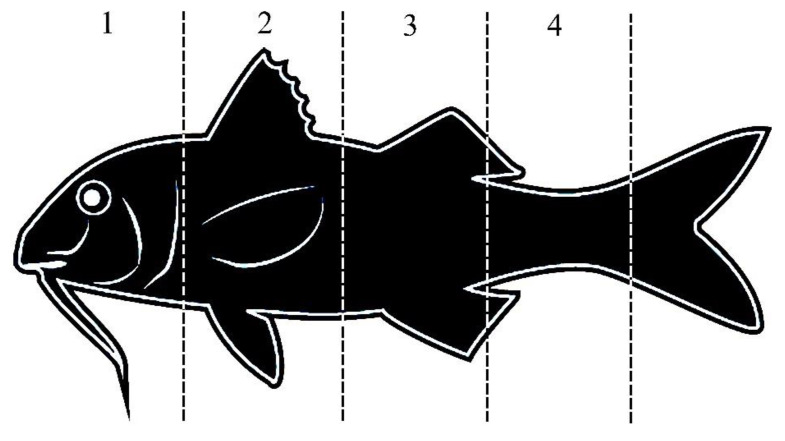
The body zones used in the present study to assess the scale damage and injuries of the *Mullus barbatus* individuals.

**Figure 7 biology-12-00649-f007:**
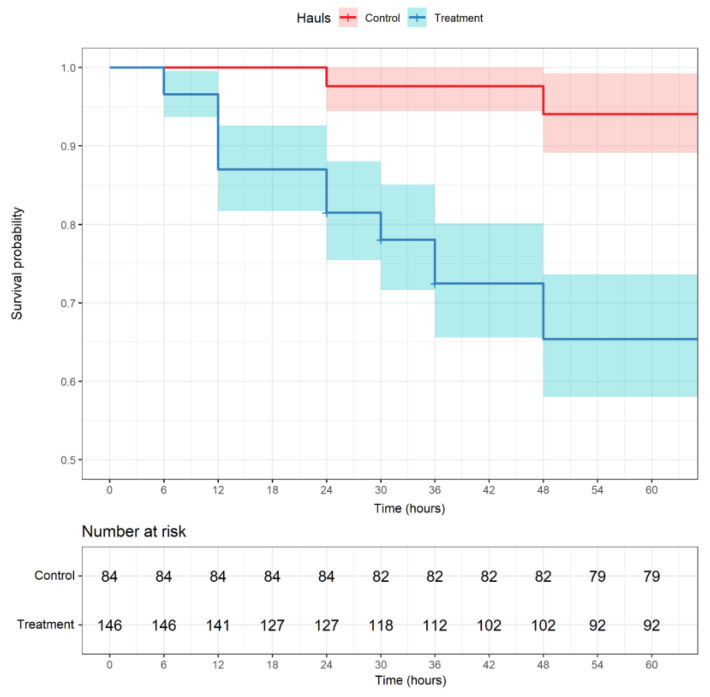
Kaplan–Meier plot showing the probability of survival of the *Mullus barbatus* individual for treatment and control hauls. Number at risk: the number of fish dying over time in both treatment and control hauls. No fish died after 48 h in either the control or treatment cages.

**Figure 8 biology-12-00649-f008:**
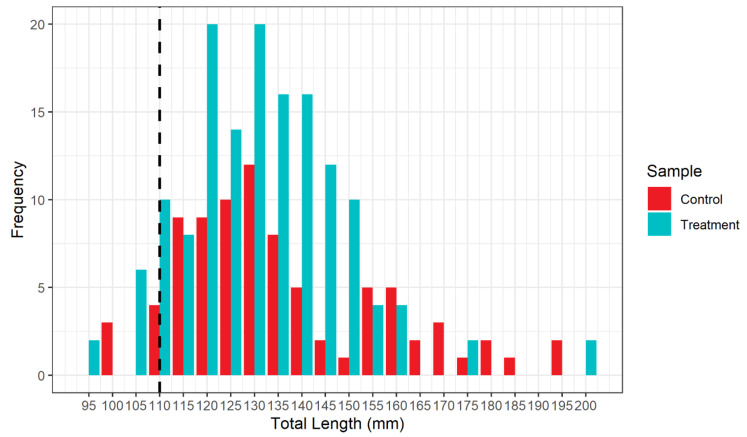
Length frequency distribution of *Mullus barbatus* sampled during the survey from treatment and control hauls. The vertical black dashed line denotes the MCRS defined by Reg. EU 1241/2019.

**Figure 9 biology-12-00649-f009:**
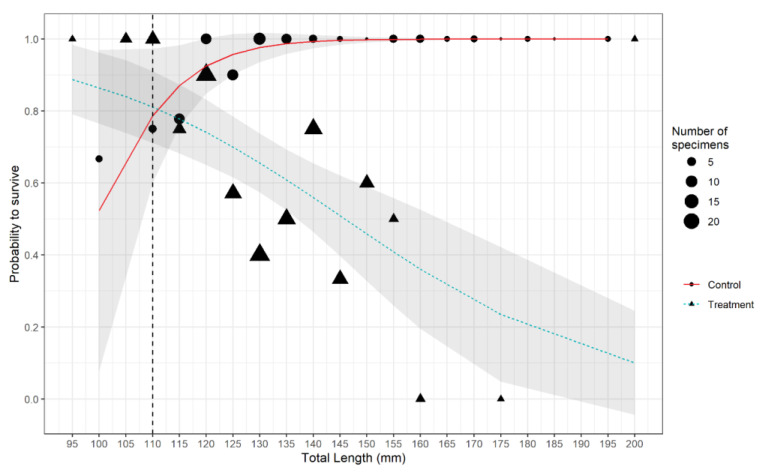
Probability of survival per class size of the individuals caught during the (A) treatment and control (B) hauls. Grey areas represent the 95% confidence intervals; the vertical black dashed line denotes the MCRS defined by Reg. EU 1241/2019.

**Figure 10 biology-12-00649-f010:**
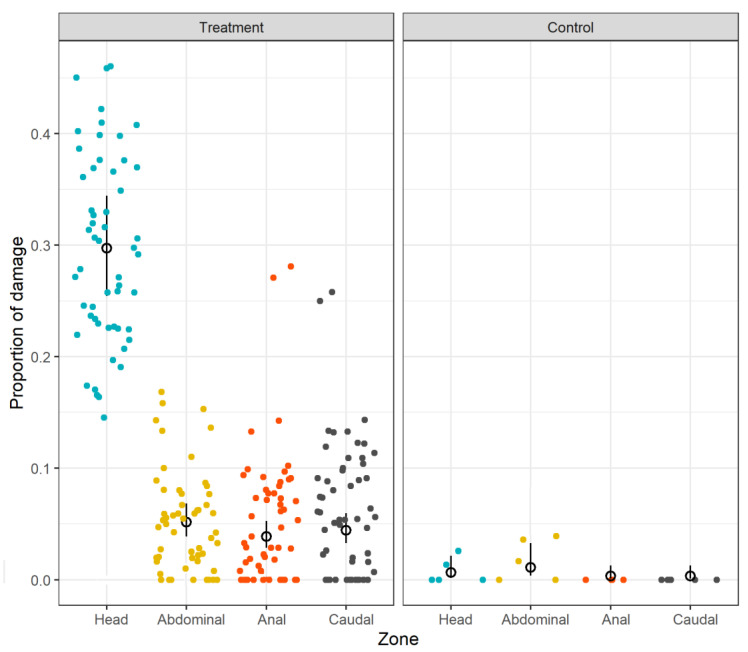
Proportion of skin damage (%) to red mullet (*Mullus barbatus*) in different body zones in the treatment and control hauls. Filled dots are the observed damage proportion for individual fish, black open circles are the mean GLMM predictions, and the bars are the 95% confidence intervals.

**Figure 11 biology-12-00649-f011:**
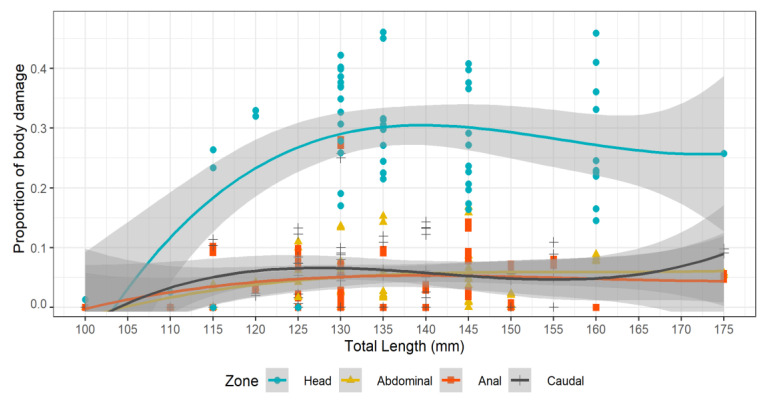
Proportion of skin damage to dead red mullet in the treatment group per class size and body zone. Trend lines indicate linear model fitting. Grey areas are 95% confidence interval.

**Table 1 biology-12-00649-t001:** Number of survivors and dead individuals by species caught in treatment (closed codend) and control (open codend) hauls.

	Control			Treatment		
Species	Dead	Survivors	Total	Dead	Survivors	Total
*Arnoglossus thori*	//	//	//	3	0	3
*Bothus podas*	1	0	1	//	//	//
*Dactilopterus volitans*	//	//	//	1	0	1
*Microchirus variegatus*	//	//	//	1	0	1
*Mullus barbatus*	5	79	84	54	92	146
*Pagellus acarne*	//	//	//	0	3	3
*Pagellus erythrinus*	//	//	//	0	5	5
*Serranus cabrilla*	//	//	//	1	2	3
*Serranus hepatus*	//	//	//	0	4	4
*Spicara flexuosa*	//	//	//	0	1	1
*Synodus saurus*	//	//	//	4	0	4

**Table 2 biology-12-00649-t002:** Results of GLM of *Mullus barbatus* survival. Std. error: standard error; TL: total length; Resid. df: residuals degree of freedom; Resid. Dev: residual deviance.

Coefficients	Estimate	Std. error	z Value	*p*-Value	Resid. df	Resid. dev	*p*-Value
Intercept	−11.975	5.789	−2.068	0.039			
TL	0.121	0.050	2.410	0.016	31	63.450	0.025
Haul_type Treatment	17.892	6.007	2.978	0.003	31	93.010	<0.001
TL: haul_type Treatment	−0.161	0.051	−3.132	0.002	30	58.405	<0.001

**Table 3 biology-12-00649-t003:** Results of GLMM of *Mullus barbatus* scale damage and injuries. Std. error: standard error; TL: total length; Df: degree of freedoms; logLik: logarithm of Likelihood; Chisq: chi square; Chi df: chi degree of freedom.

Coefficients	Estimate	Std. Error	z Value	*p*-Value	Df	LogLik	Deviance	Chisq	Chi df	*p*-Value
Intercept	−0.860	0.107	−8.001	1.23^−15^						
Zone Abdominal	−2.056	0.159	−12.893	<2^−16^	11	587.07	−1174.2	0	0	<0.001
Zone Anal	−2.353	0.168	−13.991	<2^−16^						
Zone Caudal	−2.212	0.166	−13.328	<2^−16^						
Haul_type Control	−4.182	0.636	−6.578	4.76^−11^	11	587.07	−1174.2	0	0	<0.001
Zone Abdominal: haul_typeControl	2.587	0.636	4.068	4.74^−05^	11	587.07	−1174.2	17.878	3	<0.001
Zone Anal: haul_type Control	1.675	0.670	2.503	0.012						
Zone Caudal: haul_type Control	1.534	0.669	2.293	0.022						

## Data Availability

Not applicable.

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
