# Peer review of "Escape Survival and Scale Damage Assessment of Red Mullet (Mullus barbatus Linnaeus, 1758) during Bottom Trawling in the Central Mediterranean Sea"

_biology, 2023, doi:10.3390/biology12050649_

Round 1

Reviewer 1 Report

The ms investigates the likelihood of survival from the fishing net for the red mullet (Mullus barbatus) caught in Mediterranean bottom trawl mixed fisheries. Survival of fish escapes is a critical issue that has received insufficient research attention in Mediterranean fisheries and this contribution is very relevant. The results show a surprisingly high survival of red mullet (63%) escaping from trawl nets, which is encouraging. However, the study also highlights that a substantial part of the individuals escaping through the codend mesh (37%) will likely die. This rate of “unaccounted mortality” is sufficiently high to introduce bias in the mathematical models of stock assessment, which is an important result with implication for fisheries scientific advice. The results of this research are of immediate scientific and policy interest, in particular the results are relevant to the implementation of the European Common Fishery Policy (CFP).

Author Response

Many thanks for your compliments. We hope that our manuscript will contribute to a better estimate of fishing mortality and more suitable management measures for the conservation of this important fishery resource

Reviewer 2 Report

Good paper in widely neglected area of the fisheries science. Please consider advice given. Some minor changes suggested (see attached file).

Author Response

Good paper in widely neglected area of the fisheries science. Please consider advice given. Some minor changes suggested (see attached file).

Authors: Many thanks for your compliments and comments which surely improved the quality of the manuscript. The answers to your questions are shown below.

Line 22-23: Control = not subject to the gear; treatment = subject to the gear? Please explain.

Authors: Yes, you are right, Control = not subject to the gear; treatment = subject to the gear;

we rephrased it as follows: “The survival rate of the control individuals (open codend) was higher compared to the treatment individuals (closed codend) with larger treatment fish having a higher probability of dying, while the opposite was observed in the controls.”

Line 22-23: Unclear meaning. Is this larger fish as opposed to smaller fish after trawling; and in controls smaller fish as opposed to larger fish? Please clarify.

Authors: To make the message clearer, the sentence has been rewritten as follows: “Larger fish in the treatment group had a higher probability of dying, while the opposite was observed in the controls”.

Line 24: What means "more"? More frequently? More wounds per fish? Critical wounds limiting survival?

Authors: We mean “More wounds per fish”.  Now rewritten as: “treatment fish were had significantly more wounds per fish than control fish, and where were mainly injured around the head”.

Line 26-28: I do not understand this sentence (context suggests otherwise?).

Authors: No, context does not suggest otherwise. Given that, for comparison among areas, data are very scanty about this issue before being applied to the stock assessment, something similar to a precautionary approach has been suggested.

Line 33: Fish normally are killed after the bottom trawl is hauled on deck. You have to briefly explain how this survival is effected (tub attached to a codend?).

Authors: The following sentence was added: Fish escaping from the trawl codend were collected in a detachable cage, which was lined to reduce water flow and protect the sampled fish from further fatigue and injury”.

Abstract should better explain aim, procedure and results. It is not self explanatory at the moment without reading the whole text of the paper.

Authors: Following your previous comments a new sentence about the sampling methodology has been added, which we hope better explains the procedure.

How the cage affects the flow? This should have been measured because cage does not appear in the commercial gear. At least, discuss the issue.

Authors: The cover designed was based on a system developed by Breen et al (2007), who conducted extensive flume tank and full-scale flow measurements to demonstrate that their designed had minimal effect on water flow in and around the codend, while providing the necessary protection from flow to the sample of escaping fish (see lines 133-135).  Furthermore, in this experiment  the cage and the liner were applied in both hauls/gears (control and treatment). The only differences between hauls were that the treatment haul was carried out with a closed codend while the control one was with an open codend.

Both should be 15 min

Authors: Thank you for this comment, you are right but as was already written in the sentence below “The main reason for limiting the towing duration of the control to 15 min was to avoid over-filling the sampling cages with fish. The difference in towing duration between the treatment and control should have a minimal effect on the samples of escaped fish, because the cage lining specifically protects them from fatigue after escape.” and many already published research articles on this topic applied this methodology.

Any barotrauma effects? Discuss this please.

Authors: There were no observed barotrauma effects. The deliberately slow hauling of the trawl net and hence decompression of the fish samples, as well as the relatively shallow waters in which the fishing took place, are likely to have mitigated any decompression-related barotrauma. A new sentence in the result section has been added: “No barotrauma related injuries were observed.” (Line 300)

I suspect significantly different length distribution for "control" and "treatment". Discuss this please. Your conclusion just below this note may be affected by this difference. Also, logically, fish which squeezed through should be smaller than "controls"? "Treatment" had the smallest and largest fish!

Authors: We applied the Kolmogorov-Smirnov Test on the numbers of individuals per size classes between sampled populations and no significant differences were detected between the treatment and control length distributions. As for the second question, most of the catch was composed of adults in both hauls, in addition, what you see in Figure 2 are the absolute frequencies (i.e. the numbers of individuals) if we considered the relative frequencies (or proportions) in the treatment cage there would be more undersized than in the control while, apart from some small exceptions (2 individuals in 200 mm TL), it would be the opposite for adults.

Proportion of damage - OK, but a survival rate?

Authors: The damage assessment was carried out only on dead individuals. In addition, the overall survival rate is given in lines 237-238 while Figures 7 and 9 are shown respectively the survival rate over time and the survival probability per class size.

Line 302: Fish was caught by net and not by treatment. Please rephrase.

Authors: The classification of the scale damage and injuries was removed in the revised version of the manuscript

Line 338: Logical consequence: larger fish squeeze-through = more damage.

Authors: Thank you for highlighting this. It may appear to be a logical consequence, but is not that obvious because larger fish may be more resistant to injury and associated stress. Furthermore, several studies have demonstrated negative relationship between escape mortality and injuries and individual length.  That is, smaller escaping fish have a higher likelihood of mortality and injury (see Suuronen, 2005 for review). In light of this, we have taken the opportunity to redraft this part of the discussion to make our point clearer. See lines

Line 387-389: Conclusion in the first sentence was not discussed in detail in the Discussion (as it should), therefore it is not proven.

Authors: On reflection, we agree and based on this and other comments have substantially reworded the conclusion, as follows:

This study has described an effective method for estimating escape mortality in red mullet escaping from trawl codends, in particular the use of a cover line to protect fish from injurious water flows during sampling. The observed survival rate of M. barbatus in the treatment group was 63% (55-70% CI), while the control group was 94% (87-97% CI).  It is recommended that further studies using methods like those described here should be conducted to provide valid scientific justification for the implementation of regulations trawl codend mesh size and MCRS in the Mediterranean (as defined by the Reg. EU 1241/2019). These studies should have sufficient replicates to provide a robust description of the natural variation in escape survival rates, as well as attempt to replicate variations in commercial fishing operations, including: seasonality, fishing depth, towing duration and associated catch sizes and compositions. The data resulting from this work will be essential for removing biases to fishing mortality estimates because of the inherent assumption in most stock assessment models that all es-caping fish survive.  Furthermore, it will enable fisheries managers to make more informed decisions about the implementation of different gear selectivity scenarios, based on more reliable projections of yield and biomass in the exploited populations.

Reviewer 3 Report

Review of “Escape survival and scale damage assessment of red mullet (Mullus barbatus Linnaeus, 1758) during bottom trawling in the 3 central Mediterranean Sea”. 

I have minor’s comments below, 

The introduction seems a long abstract and the end part a discussionish. Should be improved

Line: 61: Put the species' scientific name.

Line: 65: Put the species' scientific name.

The map is confusing. It needs to be improved for non-Europeans to understand. 

The work is interesting with promising results.

However, despite the promising results, I missed a better discussion about the potential use of the work in the management and conservation of the species.

In the abstract and conclusion, the issue of fish stock assessment is mentioned, however, this is never addressed in the discussion. How might job results affect inventory valuations?

I believe that inserting these small topics into the discussion could increase the value of the work.

Author Response

Comments and Suggestions for Authors

Review of “Escape survival and scale damage assessment of red mullet (Mullus barbatus Linnaeus, 1758) during bottom trawling in the 3 central Mediterranean Sea”.

I have minor’s comments below,

Authors: Many thanks for your compliments and comments, the answer to your questions are shown below.

The introduction seems a long abstract and the end part a discussion. Should be improved

Authors: Done, the Introduction has been modified according to your suggestion.

Line: 61: Put the species' scientific name.

Authors: Done

Line: 65: Put the species' scientific name.

Authors: Done

The map is confusing. It needs to be improved for non-Europeans to understand. 

Authors: The map has been replaced

The work is interesting with promising results.

However, despite the promising results, I missed a better discussion about the potential use of the work in the management and conservation of the species.

In the abstract and conclusion, the issue of fish stock assessment is mentioned, however, this is never addressed in the discussion. How might job results affect inventory valuations?

I believe that inserting these small topics into the discussion could increase the value of the work.

Authors: Thank you for your useful suggestion. To address this, some sentences about the potential use of the work in the management and conservation of the species have been added to the discussion. See lines 381-392.

Reviewer 4 Report

Figure 1 needs to present in a box the area of the Mediterranean in which the study was carried out.

Figure 2 needs to improve its quality.

Author Response

Figure 1 needs to present in a box the area of the Mediterranean in which the study was carried out.

Figure 2 needs to improve its quality.

Authors: Many thanks for your compliments, all of your suggestions have been implemented.